# Spikyball Sampling: Exploring Large Networks via an Inhomogeneous Filtered Diffusion

**Benjamin Ricaud \*** **, Nicolas Aspert and Volodymyr Miz**

LTS2, EPFL, Station 11, CH-1015 Lausanne, Switzerland; nicolas.aspert@epfl.ch (N.A.);
volodymyr.miz@epfl.ch (V.M.)
**\*** Correspondence: benjamin.ricaud@epfl.ch

**Abstract:** Studying real-world networks such as social networks or web networks is a challenge. These networks often combine a complex, highly connected structure together with a large size. We propose a new approach for large scale networks that is able to automatically sample user-defined relevant parts of a network. Starting from a few selected places in the network and a reduced set of expansion rules, the method adopts a filtered breadth-first search approach, that expands through edges and nodes matching these properties. Moreover, the expansion is performed over a random subset of neighbors at each step to mitigate further the overwhelming number of connections that may exist in large graphs. This carries the image of a "spiky" expansion. We show that this approach generalize previous exploration sampling methods, such as Snowball or Forest Fire and extend them. We demonstrate its ability to capture groups of nodes with high interactions while discarding weakly connected nodes that are often numerous in social networks and may hide important structures.

**Keywords:** networks; data over networks; snowball sampling; large scale

---

## 1. Introduction

Exploring large networks and analysing the activity within them is of crucial importance. For example, social networks have become a central source of information for citizens and have a large impact on society. A better understanding of the interaction mechanisms between users and better tools for the analysis of the activity within social networks are required. However, in practice, when exploring social networks, researchers are faced with several challenges. Firstly, the network has an overwhelming size and a high density of connections. Secondly, data collection from social networks, when not explicitly forbidden, is often restricted by APIs that limit the number of queries and their versatility. Thirdly, there is an extremely large number of users that do not bring information that is relevant to solving a research problem at hand. For instance, those users may be inactive, weakly connected, or just follow others and do not interact with the rest of the network. Such users introduce noise to collected datasets and mask the informative activity of studied social networks. Therefore, a trivial uniform sampling approach is ineffective because it does not take into account various attributes of the users and collects a lot of irrelevant data.

In order to deal with these challenges, we propose a new, general and flexible sampling method based on exploration rules. Our sampling approach takes into account nodes and edges properties. The method enables the collection of certain types of nodes and edges based on their attributes. For instance, it can focus on hubs or influencers in social networks and active users that follow them while ignoring others. This technique is a generalization and extension of the snowball sampling method [1]. Instead of having a ball expanding uniformly around initial nodes, our method, called the *Spikyball*,

1.　　reduces the expansion to a subset of the possible neighbors,
2.　　chooses this subset according to predefined rules.

This is illustrated on Figure 1. Such an approach has a number of benefits. First, exploring a selected subset of connections unlocks the exploration of large, highly connected graphs to distances of several hops. Second, the constrained size of the random subset reduces the number of requests that might need to be made to the API of a social network. Finally, guiding the exploration by rules enables a more efficient exploration, since we collect edges and nodes that are relevant to the problem and discard the rest. In an example of application discussed in this work, the nodes of interest are social network influencers and active spreaders of the information who often have a large number of followers. Hence, to collect the hubs and discard weakly connected users, we present rules that prioritize the degree of nodes. Starting from one or several initial nodes, the exploration iteratively follows the edges of the graph and expands along them. The subset of edges followed at each step makes it look like a spiky ball rather than a snowball, hence its name.

The first contribution of the present work is to show that Spikyball generalizes the family of exploration-based sampling schemes and connects the members of this family, such as Snowball sampling, Forest Fire sampling [2] or graph-expander sampling [3].

Secondly, our generalization introduces a novelty to the aforementioned approaches. Going beyond their common goal of providing a reduced but faithful representation of a full graph, our sampling approach is tunable and can direct the collection toward nodes or edges sharing a particular property within the network, be it a high degree, or some attribute associated to them. To be more concrete and motivate the choice of functions and parameters, we refer to an example of sampling a social network. In this framework, the goal is to collect the influencers (high degree nodes) among the overwhelming number of weakly connected users. The approach is general and could find applications beyond the exploration of social networks, for instance, in large graphs where the task is to capture a few key nodes that remain otherwise hidden due to the massive size of the network. Indeed, our method departs from the usual objective of graph sampling and unlocks the exploration of *regions* within large, highly connected graphs. Most of the studies on graph sampling focus on the faithfulness of the sampled graph compared to the original one, in terms of global graph properties (degree distribution, clustering coefficient, communities, etc.). Under such metrics, a good representation usually requires to collect at least 20 to 30% of the graph [4], which is prohibitive in real social networks. Our approach is different: our goal is either to collect a faithful representation of the subset of key nodes (e.g., with high degree) or to get a faithful *neighborhood* of a given node or a group of nodes and not of the entire graph. Doing so greatly reduces the amount of data to process while providing essential information about the chosen subset or region of the graph. The obtained sampled graph is especially useful for visualizing information: sampling strategies have an important impact on the visualization of a graph [5] and high degree nodes are essential for a good quality of visualization.

Thirdly, we provide an in-depth analysis of the Spikyball parameters and their impact on the sampling of synthetic and real networks. We also demonstrate the robustness of the sampling: even though the exploration of the neighborhood is partly random, it collects with high probability the key influencers around a group of users. In the case of a real social network, we show that hubs (with degree larger than 20) are sampled with a high probability, by more than 80% of the runs we performed and even 100% for nodes with degree higher than 100. Eventually, we show that some Spikyball parameters have no effect on the sampling of artificial random networks, while having one in the case of real networks. We suggest new uses of these samplings for assessing the structure of a real network by comparing the degree distribution obtained with different parameters.

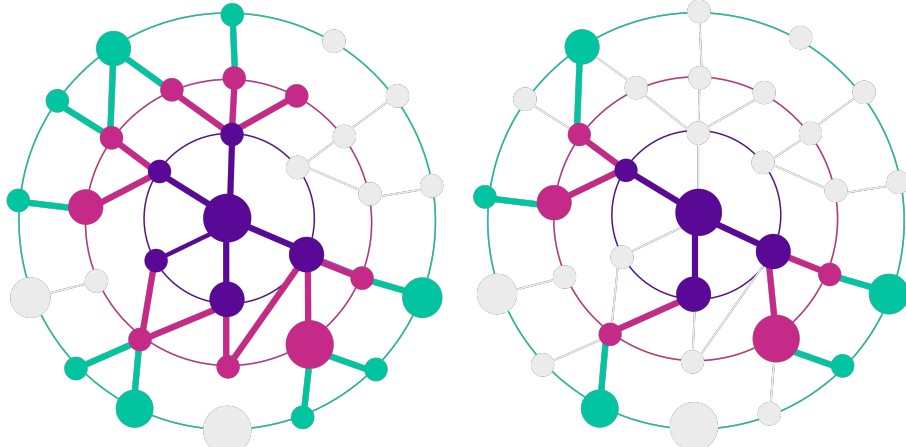

**Figure 1.** Snowball (**left**) and Spikyball (**right**) sampling example. The sampled nodes are colored in purple, pink, and green, the non-sampled ones are in grey. Starting from the central node (user defined), both samplings expand in successive hops, following neighbor connections (purple circle is 1-hop, pink circle is 2-hop, and green circle is 3-hop). The size of the nodes symbolizes their importance and the spikyball focus on collecting them in priority. This importance can be related to their degree or to some other attribute associated to them.

## 2. Related Work

There are two main families of graph sampling methods adopting a neighbor exploration strategy. The first family is based on random walks (RWs). Examples are the re-weighted random walk, the Metropolis-Hasting random walk [6], CNRW and GNRW [7], CNARW [8] or Frontier Sampling [9]. The general principle is to guide random walks with probabilities assigned to edges. These probabilities define which neighbors to visit as the exploration progresses. In the re-weighted and Metropolis-Hasting RWs, the goal is to reduce potential biases toward high degree nodes and obtain a faithful reduced graph. The assessment criteria are degree distribution and graph properties such as diameter or clustering coefficient. For the CNARW, rules are introduced to push the exploration to go farther at each step, favoring a faster sampling of the graph. Frontier Sampling uses multiple RWs in parallel.

The second family of graph crawling algorithms starts from an initial node or group of nodes and expands around it either with a breadth-first search (BFS) or depth-first search (DFS) approach. Examples are Snowball sampling [1,10], where close regions around the initial starting point are densely sampled while regions that are located further away from it may not be sampled at all. Moreover, in large, densely connected, networks, snowball gets quickly surrounded by the overwhelming number of neighbors. Forest Fire [2] and the Expander graph approach [3] provide solutions for large networks. These approaches achieve scalability by following a reduced random subset of the edges during the expansion. The Spikyball approach unifies these latter samplings into a unique framework and adds new exploration options. Its tunable rules enable the preferential collection of nodes and edges with specific properties.

More precisely, what distinguishes the methods within the second family are the spreading rules. The snowball sampling is a breadth-first search with a stopping parameter. At each iteration, the sampling propagates to all the neighbors and is stopped after $N$ iterations. For the Forest Fire, the propagation takes place on a subset of the edges selected randomly. For each newly burned (or explored) node, a number $x$ of its neighbors is selected to spread the "fire" further. This random number follows a geometric distribution and is independent of the number of neighbors. This independence decreases the influence of highly connected nodes (possessing many edges where the fire could propagate). This is desirable in the context of graph sampling when the sampled graph must have a degree distribution as close as possible to the one of the full graph and Forest Fire is designed for this purpose [11]. However, this random

selection may fail at collecting a set of nodes with particular properties that are considered important for the research problem at hand. In the Expander graph method [3], the subset of neighbors to spread to is selected according to the target node connections. The nodes giving the largest increase in the number of neighbors of the sampling are collected. The rationale is to maximize the chance to collect new communities. However, estimating the optimal nodes to spread to needs costly requests. For each candidate, all its connections have to be requested and this can result in a computationally intractable solution when dealing with applications that involve social media. Furthermore, this also leads to a partial sampling of the core nodes of a community. Indeed, non-sampled nodes with strong connections to the already sampled nodes, even with high degree, may not be collected. In the Rank Degree method [12], the exploration rule is slightly different, the degree of the neighbors guides the propagation. Furthermore, the initial set may contain many nodes taken at random, while the previous sampling uses a single or a few starting points.

Studies on graph sampling assess the quality of the sampling using various criteria from the degree distribution shape, topological properties such as diameter, clustering coefficient [11,13,14] to spectral properties of the graph Laplacian [15]. These criteria are highly relevant for general graphs. However, in the particular case of social networks, other constraints on the sampling process are of central importance. Firstly, the collection of influencers, spreaders and other active users may be more important than having a faithful representation of the graph in terms of diameter or clustering coefficient. Secondly, a large number of users mask the important activity without bringing much relevant information (apart from highlighting its virality). Thirdly, the collection via proprietary APIs provides reduced access to the network and it is a key factor limiting the sampling. The performance of RWs explorations under this constraint has been studied in [16], showing important discrepancies between approaches. In [17], a well-chosen initial set of users, related to the topic of interest (e.g., a political event or a technology topic), improved the results of the Forest Fire sampling, allowing to follow and measure the diffusion of information. The Spikyball carries the same idea but extends the focus from the initial group of users to the exploration rules. The sampling biases of existing sampling schemes are studied in [18] and are shown to have some benefits for particular applications, for example, discovering new communities. This is an additional argument toward the utility of new exploration approaches having flexible exploration rules, such as ours.

## 3. Proposed Method

In this section, we first describe the general method. We then explain in details the exploration rules in Section 3.1, with the different parameters and their influence on the sampling. We introduce names for samplings with particular parameter values, to better distinguish them in the rest of the paper. Finally, we show the connection with the main sampling methods found in the literature in Section 3.2.

Let $\mathcal{L}_0$ denote the set of initial nodes from where the sampling starts. This is the initial layer. The algorithm is based on the exploration of the graph in successive layers $\mathcal{L}_1, \mathcal{L}_2, \mathcal{L}_3 \ldots$, built around the initial one and expanding along the graph edges. Each layer is a set of nodes, that have been obtained from the expansion process applied at the previous layer. Different parameters set the expansion rules, selecting subsets of edges to follow, from one layer to the next. The expansion rules are the crucial ingredient that determines the final properties of the sampled graph. The main algorithm is shown on Algorithm 1. In the case of an attributed graph, the exploration may depend on the graph structure as well as the properties associated to the nodes and edges of the graph.

---

**Algorithm 1:** Spikyball algorithm.

    **Data:** Initial set of nodes $\mathcal{L}_0$, number of layers $K$, graph to sample $G$
    **Result:** A graph $G_s$ sampled from $G$
    initialization $\mathcal{L}_k \leftarrow \mathcal{L}_0$, $G_s \leftarrow \varnothing$, $\mathcal{L}_T \leftarrow \mathcal{L}_0$;
    **for** $k \leftarrow 0$ **to** $K-1$ **do**
        |  nodes-info $\leftarrow$ GetNodesInfo $(\mathcal{L}_k)$;
        |  $E_k \leftarrow$ GetNeighbors $(\mathcal{L}_k)$;
        |  $E_k^{(\text{in})}, E_k^{(\text{out})} \leftarrow$ FilterEdges $(E_k, \mathcal{L}_T)$;
        |  $\mathcal{L}_{k+1}, E^{\text{sampled}} \leftarrow$ SampleEdges $(E_k^{(\text{out})}$, nodes-info$)$;
        |  $G_s \leftarrow$ AddToGraph $(G_s, \mathcal{L}_k, E_k^{(\text{in})}, E^{\text{sampled}}$,nodes-info$)$;
        |  $\mathcal{L}_T \leftarrow$ Union$(\mathcal{L}_T, \mathcal{L}_{k+1})$;
    **end**

---

The first two functions of the algorithm access and collect the information from the network. The function GetNodesInfo is the one retrieving the information associated to the nodes within each layer $\mathcal{L}_k$. This information may be required when dealing with an attributed graph, in the case where the sampling is guided by the attributes. While this algorithm concerns the general case of graphs having attributes, it can be used without modifications when no attributes are present. In this case, the GetNodesInfo will always return an empty set. The function GetNeighbors collect the edges incident to the nodes in $\mathcal{L}_k$ as well as the data attached to these edges if available. Both functions (GetNodesInfo and GetNeighbors) will make $|\mathcal{L}_k|$ requests, with $|\mathcal{L}_k|$ being the number of nodes in $\mathcal{L}_k$. In many cases, for graphs without attributes or for some networks allowing to join node and edge requests, GetNodesInfo is not needed and the number of requests will be divided by 2. We assume that the information about the neighbors is not collected, except their node id which is encoded in the edges. Indeed, collecting information from the neighbors would require an additional query for each neighbor, which can be quickly prohibitive and limited in the case of a social network.

When the raw data has been collected from the network, it is further processed by three different functions. The edges are sorted by FilterEdges between edges connecting the source nodes to nodes already collected in previous layers $E_k^{(\text{in})}$ and the edges pointing to new nodes $E_k^{(\text{out})}$. Furthermore, this function can be used to remove edges according to some criteria, for example, if the weight of an edge is smaller than a given value set by the user. The function SampleEdges selects the edges to follow from the set provided by FilterEdges and outputs the nodes that will form the new layer, ready for the next exploration step. The exploration rules are encoded inside this function and are explained in more details in Section 3.1. To perform the sampling, SampleEdges can take into account the data collected from the nodes in $\mathcal{L}_k$ and the data associated to the edges. The last function, AddToGraph, adds the new nodes and their connections (possibly with their attributes) inside $G_s$. Eventually, the union of two sets is performed with Union to update the list of sampled nodes $\mathcal{L}_T$.

*3.1. Exploration Rules*

The edge sampling rules are encoded in the function SampleEdges which takes 2 input arguments. The first one is the list of edges with their properties, $E_k^{(\text{out})}$, that connect nodes from $\mathcal{L}_k$ to their neighbors not already sampled (not in $G_s$). The second one is the data associated to the nodes in $\mathcal{L}_k$ that may influence the selection of the edges. Within the function is performed a selection of edges to explore, among the one in $E_k^{(\text{out})}$. The target nodes of the selected edges will be the elements of $\mathcal{L}_{k+1}$, to explore in the next step $k+1$. Several exploration schemes can be defined with different sets of rules and different properties. The key element is a probability mass function $p_k$ associated to the set of edges $E_k^{(\text{out})}$ that guides the choice of edges to follow from the layer $k$ to the next. This can also be seen as a conditional probability $p_k(j|i)$ of choosing $j$ at layer $k+1$ if $i$ has been collected at layer $k$.

For the Spikyball, this probability mass function depends on the properties associated to the edge and its source and target nodes. The influence of these properties can be tuned and lead to different propagation schemes. We introduce three real numbers $\alpha, \beta, \gamma$ and three functions $f, g, h$. These functions map the feature space of the source node, edge and target node respectively to positive real numbers. We have, at layer $k$,

$$p_k(e_{ij}) = p_k(j|i) = \frac{f(i)^\alpha g(i,j)^\beta h(j)^\gamma}{s_k}, \tag{1}$$

where $s_k$ is the normalization depending on the number of nodes at layer $k$ and their neighbors $\{\mathcal{N}(i)\}_{i\in\mathcal{L}_k}$:

$$s_k = \sum_{i\in\mathcal{L}_k}\sum_{j\in\mathcal{N}(i)} f(i)^\alpha g(i,j)^\beta h(j)^\gamma. \tag{2}$$

The normalization varies with the layer as there is a different number of nodes and neighbors at each step, however, the mappings from features to positive numbers are independent of the layer. In some cases, the features may depend on the layer. For example, the number of connections of a node that are not connecting it to nodes already collected in previous layers (see below).

There are many possibilities for defining $f, g, h$ and the exponents. In the present work, we focus on a few cases that are representative of the general approach, and of potential interest for the exploration of social networks. In the following, we denote by $w_{ij}$ the weight associated to edge $e_{ij}$ between node $i$ and node $j$. The weighted degree of node $i$ is $d_i$ and $d_i^{(out)}$ is the sum of weighted connections from node $i$ to nodes of $G$ not in $G_s$. We also introduce $d_j^{(in)}$, the number of connections of node $j$ from nodes in layer $k$. Note that $d_i^{(out)}$ and $d_j^{(in)}$ both depend on layer $k$, although omitted from the notation for simplicity. The number of edges in $E_k^{(out)}$ is denoted $N_k = |E_k^{(out)}|$.

**Spikyball:** This is the general setting. A random subset of the edges is selected at each layer $k$ following the probability mass function of Equation (1). If the graph is attributed, the probability to pick an edge can depend on the data, on the edge, and on the connected nodes via the functions $f, g, h$. For example, in a social network, one may want to sample more edges from highly active users. The number of posts of these users could be the value (node data) influencing the probability to sample the edge. If the graph does not have any data associated to the nodes or edges, the functions take the natural properties of edges (weight) and nodes (degree) which reduces the probability function to:

$$p_k(e_{ij}) = \frac{d_i^{(out)\alpha} w_{i,j}^\beta d_j^{(in)\gamma}}{s_k}. \tag{3}$$

**Uni-edge ball:** The edges are selected uniformly at random without replacement, $\alpha, \beta, \gamma$ are zero and

$$p_k(e_{ij}) = \frac{1}{s_k} = \frac{1}{N_k^{(out)}}. \tag{4}$$

Optionally, the selection may take the edge weights into account by defining a probability mass function proportional to the weights, $\beta = 1, g(i,j) = w_{ij}$:

$$p_k(e_{ij}) = \frac{w_{ij}}{s_k}, \quad \text{with} \quad s_k = \sum_{(i,j)\in E_k^{(out)}} w_{ij}. \tag{5}$$

**Uni-node ball** or **Fireball:** In this configuration, the source nodes are selected uniformly at random without replacement, hence $\beta, \gamma$ are zero and $\alpha = -1$

$$p_k(e_{ij}) = \frac{1}{d_i^{(out)}} \frac{1}{s_k}. \tag{6}$$

As for the uni-edge ball, optionally, the selection may take the edge weights into account with $\beta = 1$ and $g(i,j) = w_{ij}$. The name Fireball refers to its similarity with the Forest Fire sampling (see next section for more detailed explanation).

**Hubball family:** This family relies on the degrees of the nodes in $\mathcal{L}_k$. The probability is the one of Equation (3) where $\beta = 1 \; \gamma = 0$:

$$p_k(e_{ij}) = d_i^{(out)\alpha} \frac{w_{ij}}{s_k}. \tag{7}$$

It is a one-parameter family that favors the selection of connections originating from hubs when $\alpha > 0$. Note that it has an influence on the outbound connections from highly connected nodes but not on the nodes themselves: the nodes in $\mathcal{L}_k$ have already been chosen and the choice is independent of the degree of the target nodes.

**Coreball family:** This sampling aims at capturing the core of the communities surrounding or including the initial nodes. This is in reference to the $k$-core measure that evaluate the hierarchy within a community and its core users. The probability relies on the degree of the target nodes. In Equation (3), $\alpha = 0$, $\beta = 1$, while $\gamma$ is kept free:

$$p_k(e_{ij}) = \frac{w_{ij}}{s_k} d_j^{(in)\gamma}. \tag{8}$$

For $\gamma > 0$, it favors the exploration of the core of a community, by collecting with a higher probability the nodes that are the most connected to the ones in layer $\mathcal{L}_k$. However, keeping a non-zero probability for connections with weakly connected nodes, the sampling gets a chance to explore outside of the already captured communities.

**Remark:** the Hubball with $\alpha = 0$, the Coreball with $\gamma = 0$ and the Uni-edge ball have the same probability distribution and are therefore the same sampling scheme. The Hubball with $\alpha = -1$ is the Uni-node ball.

*3.2. Connection to Existing Exploration Samplings*

**Snowball:** This sampling is the simplest one in terms of expansion rules. This is a particular case of the spikyball where all the edges are selected and the next layer contains all the neighbors of the nodes in $\mathcal{L}_k$. The probability mass function is not relevant here.

**Forest Fire and Fireball:** The Spikyball is close to the one performed with the Forest Fire approach [2], where the random selection of edges to burn does not depend on the degree of the node it connects to (or from). To obtain the same behavior we define a particular spikyball called Fireball. To keep the same notation to the one introduced in [2, A.1], the number of edges to select from $E_i^{(out)}$ is $n * p_f/(1 - p_f)$ where $p_f$ is the forward burning probability and $n$ is the number of nodes in the Fireball layer. In this Fireball configuration, each source node will have an equal probability to be selected as for all nodes $i$ in layer $k$, summing over its neighbors $\Omega_k(i) = \mathcal{N}(i) \cap E_k^{(out)}$ leads to a uniform probability:

$$\sum_{j \in \Omega_k(i)} p(e_{ij}) = \frac{1}{s_k} \sum_{j \in \Omega_k(i)} \frac{w_{ij}}{d_i^{(out)}} = \frac{1}{s}, \tag{9}$$

with $w_{ij} = 1$ if $i$ and $j$ are connected and zero otherwise. Notice that there may be a difference between Fireball and Forest Fire. In Forest Fire, a random number $x$ is drawn for each node $i$. If the node possesses fewer edges than this number $d_i^{(out)} < x$, all edges are selected. In this case, it acts as

a random selection with repetition. However, when $x \geq d_i^{(out)}$, it is a selection without repetition. We can not obtain this behavior with our layered approach. In practice, it should make little difference. However, on graphs where a large number of nodes with small $d_i^{(out)}$ are encountered the difference may have a visible impact.

**Expander-Graph Ball**: In this scheme [3], the nodes selected at step $k + 1$ are the ones that increase the number of neighbors of $G_s$ the most. In that case, the number of connections of the target node to nodes that are not in $G_s$, $d_j^{(out)}$ is the feature used in $p_k$. So that $\alpha = 0$, $\beta = 0$ and $\gamma > 0$. The functions $f, g$ are ignored and $h(j) = d_j^{(out)}$. If one needs to enforce the expander-graph behavior, it can be achieved by increasing $\gamma$. In order to obtain $d_j^{(out)}$, the neighbors of the neighbors of nodes in $L_k$ have to be requested and it may be prohibitive for large, highly connected, networks. Although the functions are similar to the ones of the Coreball, it differs by using ($d_i^{(in)}$ instead of $d_i^{(out)}$).

## 4. Theoretical Properties

In this section, we establish some important properties of the Spikyball family that help to understand the general behavior of these sampling methods. The influence of the parameters on the degree distribution of the collected nodes is made explicit and will be discussed in more details.

There are three main results in this section. We first prove that the degree distribution of samplings obtained with the Hubball family is independent of the parameter $\alpha$, for a large number of synthetic networks. Since this family contains the Forest Fire sampling, we prove that this latter sampling, on most random networks, will lead to an equivalent degree distribution as any Hubball defined in the previous section. The reader has to keep in mind that it may not be the case for real networks. This is due to the independence of the degree of a node with respect to the degree of its neighbors in most random networks. The second result goes further in this direction and shows explicitly the dependence of the degree distribution of the sampled graph on the relationship between neighbor nodes. Again, this is for the case of Hubballs. In real networks, the degree of neighbor nodes may be related. For example, high degree nodes may favor connections with other high degree nodes. In that case, it is more probable to find a high degree node when randomly jumping from a high degree node to one of its neighbors. Theorem 2 shows that such relation between the degree of nodes can be revealed indirectly by analysing the difference of degree distribution given by the members of the Hubball family.

The last result concerns the Coreball family and the influence of its parameter $\gamma$ on the sampling. For any graph, a large $\gamma$ will make the Coreball sample more high degree nodes than what the Snowball would do. For negative $\gamma$, more weakly connected nodes will be sampled.

For the sake of simplicity, we assume in this section the graphs to be unweighted. In order to establish the first result, we introduce a property of a graph defined as follows:

**Property 1** (P1). *The degree $d_a$ of a node $a$ is independent of the degree of its neighbors. The conditional probability $p(d_a|d_b)$ of having a node $a$ with degree $d_a$ if its neighbor $b$ has degree $d_b$ is:*

$$p(d_a|d_b) = p(d_a). \tag{10}$$

This property is shared by many random networks, where the creation of edges is independent of the degrees of both the source and target nodes. This is true for Erdős-Renyi (independent of any degree) or for Barabási-Albert graphs (independent of the degree of the source node). However, this may not hold for real networks. For example, in networks with a strong hierarchy, having a large $k$-core, high degree nodes are more connected together than to small degree nodes at the periphery.

### 4.1. Hubball Family

The first theorem is a direct consequence of Property 1.

**Theorem 1.** *The degree distributions of the nodes collected with the Hubball family for any α on a graph G with Property 1 are all equivalent.*

This result is interesting as it makes it possible to distinguish an artificial random graph (Having Property 1) from a real network by inspecting the sampled nodes with several Hubballs. It also helps understand the limits of relying on the source node degrees for sampling a graph. This effect is illustrated on Figure 2 in the experimental part. Indeed, similar degree distributions are obtained for the Hubball family (Fireball, Uni-edge ball and Hubball 2) on random graphs.

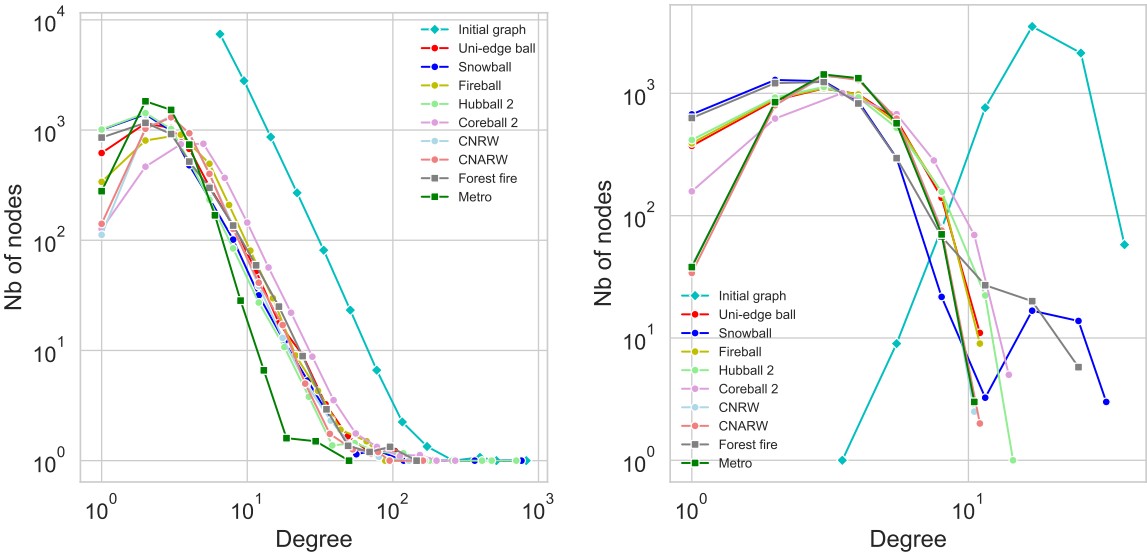

**Figure 2.** Degree distribution for different sampled graphs on Barabási-Albert (left) and Erdős-Renyi (right) random graphs. The sampling reduces the graph to 10% of its original size (50k nodes). The curve corresponding to the initial graph (not sampled) is in cyan. The different samplings provide close results. The slope is identical except for the Metropolis Hasting one, which is steeper for B-A and reaches a maximum shifted toward high degree nodes in E-R.

**Proof.** At layer $k$ of the sampling process, the neighbors of the nodes of this layer are selected according to the degree of the layer nodes: $d_i^{(out)}$ for $i \in \mathcal{L}_k$. Since the degree of neighbors is independent of the degree of the nodes in $\mathcal{L}_k$ by P1, this has no influence on the degree distribution of the selected neighbors. □

The next result is more complex and more precise regarding the action of the Hubball family over the degree distribution. Let us denote by $p(d_a)$ the probability of selecting a node with degree $d_a$, and $p(d_a|d_b)$ the conditional probability of having a node with degree $d_a$ if its neighbor has a degree $d_b$. The layer $k$ of a Spikyball sampling contains $n_k$ nodes. The degree distribution of these nodes is related to the graph and the sampling process. In order to understand more precisely this relationship, we model the evolution of the degree distribution from layer to layer. We assume that the degree of each node at layer $k$ is a random variable that has been selected from a degree distribution $q_k$ that depends on the layer. For the following results, we assume the Spikyball sampling to occur on a large graph. With a high number of nodes in the graph, the number of times a node with degree $d_a$ appears at layer $k$, with $n_k$ nodes, is well approximated by a binomial distribution. The following

Lemma creates a first relationship between the probability to select a node with a given degree with the exploration rules.

**Lemma 1.** *Assuming a large graph, the probability of selecting a node with degree $d_a$ at layer k of a spikyball sampling is given by:*

$$q_k^{(s)}(d_a) = \sum_{n=1}^{n_k} p_k(e_{aj}) d_a B(n, n_k, q_k(d_a)), \tag{11}$$

*where $B(n, n_k, q_k(d_a))$ is a binomial distribution associated to obtaining n nodes of degree $d_a$ in $n_k$ trials from the degree distribution $q_k$ at layer k. Through its normalization, $p_k(e_{aj})$ depends on n, the number of nodes with degree $d_a$ in layer k, as $s_k = \sum_{i=1}^{n_k} d_i^\alpha$.*

**Proof.** This probability $q_k^{(s)}(d_a)$ combines the probability of having a node with degree $d_a$ in $\mathcal{L}_k$ together with the probability to select it via the exploration rules. Let $S$ denote the event of selecting a node with degree $d_a$ in layer k, and $C_n$ the event: $n$ nodes of degree $d_a$ are present in the layer k. The probability $q_k^{(s)}(d_a)$ is given by the relationship:

$$q_k^{(s)}(d_a) = \sum_{n=1}^{n_k} p(S \cap C_n) = \sum_{n=1}^{n_k} p(S|C_n) p(C_n).$$

The probability $p(C_n)$ is given by a binomial distribution where the probability of a success, $q_k(d_a)$, is the probability of selecting a node with degree $d_a$. The conditional probability $p(S|C_n)$ is given by the Spikyball rules. For $C_1$, $p(S|C_1) = p_k(d_a) = p_k(e_{aj}) d_a$ is the probability to select a node with degree $d_a$. □

**Theorem 2.** *Let G be a large graph being sampled using a member of the Hubball family. Let $q_k$ be the degree distribution associated to the sampling at layer k. The number $n_k$ of nodes in layer k is assumed to be large. There exists a small $\varepsilon > 0$ such that the degree distribution of the sampled nodes at layer $k + 1$ is given by:*

$$q_{k+1}(d_a) = \sum_b p(d_a|d_b) \left( \frac{d_b^{\alpha+1}}{\bar{s}_k} n_k q_k(d_b) + \varepsilon \right), \tag{12}$$

*with $\bar{s}_k = n_k \overline{d^\alpha}$ and $\overline{d^\alpha}$ is the mean value of the degree to the power $\alpha$ when $n_k$ nodes are drawn with the degree distribution $p_k$. The bound $\varepsilon$ depends on $n_k$ and $q_k(d_b)$ and decreases with $q_k(d_b)$.*

This theorem reveals how the degree distribution of the sampled nodes are influenced by the sampling rules and by the conditional probability. If the degree of the target node $d_a$ is independent of the source node i.e., $p(d_a|d_b) = p(d_a)$, the sampling rules will not affect $q_{k+1}(d_a)$ so that all the Hubballs will give the same result. However, in the range of degrees where the probability is not independent, the sampling will change the degree distribution with an increase for high degree nodes and a decrease of weakly connected nodes when $\alpha > -1$ and conversely for $\alpha < -1$. This effect can be seen in the experiment part. It is illustrated on a particular example on Figure 3a. From it, one can deduce that $p(d_a|d_b) = p(d_a)$ for large degree nodes $d_a > 20$ as there is no change of the degree distribution for different values of $\alpha$, except maybe for $\alpha = -2$. Although differences are light in the region of weakly connected nodes, it suggests that the source and target node degrees are not completely independent. Nodes with a small degree tend to be more connected to nodes with close degree: the degree distribution shows an increase in this range as $\alpha$ decreases. In comparison, on Figure 3b, the Coreballs with different $\gamma$ have a much larger impact on the degree distribution. Even if the effect of the Hubball parameter is weak on this graph (Facebook graph), it may be much more pronounced from graphs with a high hierarchy for example, where hubs connect to hubs more than to nodes with a smaller number of connections. Measuring the difference of the Hubballs

sampling can lead to an estimate of the dependence between neighbors degrees and lead to a better understanding of the network connections.

**Proof.** At layer $k$, we have

$$q_{k+1}(d_a) = \sum_b p(d_a|d_b) q_k^{(s)}(d_b),\tag{13}$$

where $q_k^{(s)}(d_b)$ is given by Lemma 1 and $p(d_a|d_b)$ is the conditional probability defined earlier. In order to simplify the expression of $q_k^{(s)}(d_b)$, we will replace $s_k = \sum_{i=1}^{n_k} d_i^\alpha$ by $\overline{s_k}$, which is independent from $n$ the number of nodes having a degree $d_b$ in $\mathcal{L}_k$. As a consequence, the probability to select a node with degree $d_a$ in a set where $n$ nodes with degree $d_a$ are present will be replaced by $d_a^\alpha / \overline{s_k} \times n$. Assuming $n_k$ large and for small $n$, $d_a^\alpha / \overline{s_k} \times n$ is a good approximation of $p_k(e_{aj})$, since replacing a few $d_i$s by $d_b$s in $s_k$ is a small perturbation of $\overline{s_k}$, bounded by $\varepsilon/2$. For large $n$ this approximation does not hold anymore, however for $n \geq n_k q_k(d_a)$ the probability $B(n, n_k, q_k(d_a))$ decreases exponentially with $n$ (Hoeffding's inequality) and the error for large $n$ can be bounded by $\varepsilon/2$. The bound $\varepsilon$ hence decreases as $n_k$ and $q_k(d_a)$ get smaller. From (11), we can write

$$q_k^{(s)}(d_b) = \sum_{n=1}^{n_k} \frac{d_b^{\alpha+1}}{s_k} nB(n, n_k, q_k(d_b)) + \varepsilon = \frac{d_b^{\alpha+1}}{s_k} \sum_{n=1}^{n_k} nB(n, n_k, q_k(d_b)) + \varepsilon.$$

Since $B$ is a Binomial distribution, the sum over $n$ of the above expression yields $nq_k(d_b)$. □

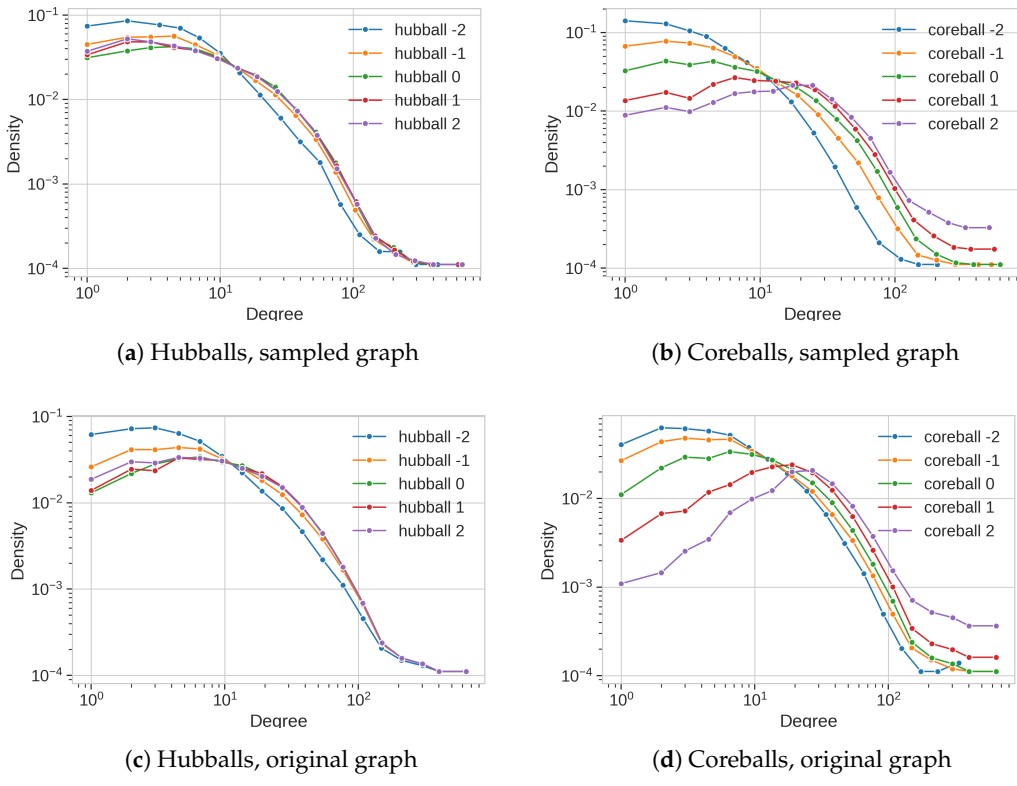

(**a**) Hubballs, sampled graph　　　　　　　　　　(**b**) Coreballs, sampled graph

(**c**) Hubballs, original graph　　　　　　　　　　(**d**) Coreballs, original graph

**Figure 3.** Degree distribution of the sampled nodes for Hubballs (**left**) and Coreballs (**right**) with different parameters, on the Facebook graph. For the sampled nodes, their degree in the sampled graph (**top**) and in the initial graph (**bottom**) lead to two different degree distributions. They are different since, in the sampled graph, edges to unsampled nodes are not present. The Hubball with $\alpha = 0$ is equal to the Coreball with $\gamma = 0$, hence plots can be compared.

*4.2. Coreball Family*

The last theoretical result concerns the Coreball family and shows the influence of the parameter $\gamma$ on degree distribution of the sampled nodes.

Let us define a well-connected graph $G$. Let $S$ be a random set of nodes of $G$. In a well connected graph $G$, the probability of having one or more direct neighbors of $S$ connected to two or more nodes in $S$ is high.

**Theorem 3.** *In a well-connected graph, the Coreball family changes the degree distribution of the collected nodes compared to the Snowball sampling with high probability. If the degree distribution changes, for $\gamma > 0$ the density of nodes with degree 1 decreases and for $\gamma < 0$ it increases.*

This result is illustrated on Figure 3b for the Facebook graph. Even if the theorem is limited to nodes with degree 1, the results hold for higher degrees as shown on the figure. The shape of the degree distribution is changed as $\gamma$ evolves, with an inflexion point around nodes with degree 20. This makes the exponent $\gamma$ of the Coreball family an effective parameter for shaping the degree distribution of the sampled nodes. The degree $d_j^{(out)}$ used in Coreball is a partial view of the real degree of node $j$. However, it is a good estimate of the real degree.

**Proof.** Let $p(i)$ denote the probability to collect node $i$ and $p(j|i)$ the conditional probability of collecting node $j$ if node $i$ has been collected. Since the collection follow the edges (except for the initial node), one can write

$$p(j) = \sum_{i \in \mathcal{N}_j} p(j|i) p(i),$$

where $\mathcal{N}_j$ is the set of neighbors of node $j$. Let us assume a subset $\mathcal{S}_j \subset \mathcal{N}_j$ of nodes have been collected at layer $k$. The probability to collect node $j$ at layer $k+1$ is:

$$p(j) = \sum_{i \in \mathcal{S}_j} p(j|i).$$

In the present work, the conditional probability $p(j|i) = p_k(j|i) = p_k(e_{ij})$ is chosen to influence the collection of some category of nodes. For Coreball:

$$p(j|i) = w_{ij} d_j^{(out)\gamma}.$$

The probability to collect node $j$ depends on $d_j^{(out)}$, i.e., the number of connection it has with nodes at layer $k$. For $\gamma > 0$ the neighbors with highest connections to layer $k$ will be collected with a larger probability. Since the real degree of a neighbor is unknown, $d_j^{(out)}$ is different form $d_j$ except when $d_j = 1$. Since the graph is highly connected, there is a high probability that $d_j^{(out)} > 1$ for some nodes, so that $p(j|i)$ will not be uniform.  $\square$

## 5. Experimental Evaluation

This section complements the theoretical one with multiple experiments and investigates different aspects of the Spikyball family. Firstly, it compares the Spikyball to the main sampling methods found in the literature. Secondly, the effect of the Spikyball parameters on the sampling is analysed on a practical case. We identify some tasks where it is particularly interesting to use a Spikyball: when the sampling needs to focus on the high degree nodes. Thirdly, its robustness and capacity of sampling high degree nodes (social network "influencers") is evaluated.

*5.1. Comparing to the Literature*

In order to compare the Spikyball approach to the methods presented in the literature, we investigate their quality as graph sampling methods. We first focus on the degree distribution of the sampled graphs. Using the open-source graph sampling toolbox [19], we compare our main Spikyball variants, to the Snowball, Forest Fire, CNRW and CNARW and Metropolis-Hasting sampling implemented therein. The results of sampling random networks are shown in Figure 2 and on a real network in Figure 4. The real network is a part of the Facebook graph, provided in the toolbox [20]. There is a general decrease in the number of nodes for each degree value, which is of course due to the fact that we subsample the network (to 10% of its original size). As explained in Section 3, the Fireball and Forest Fire give similar results as their random exploration is almost identical. Both methods are close to the Snowball sampling distribution. As expected, the Coreball 2, tends to collect more high degree nodes than the other methods. This is more pronounced in the real network case. On random networks, the pink curve is below the others for degrees smaller than 5 and slightly above beyond this value. The Metropolis Hasting method, which does not belong to the same sampling family, collects more low degree nodes. It is the closest to the shape of the initial graph distribution for the real network and the Erdős-Renyi one. However, it is one of the worst for the Barabási-Albert one with a steeper slope. All methods have difficulties to capture nodes with extreme degrees (too small or too large).

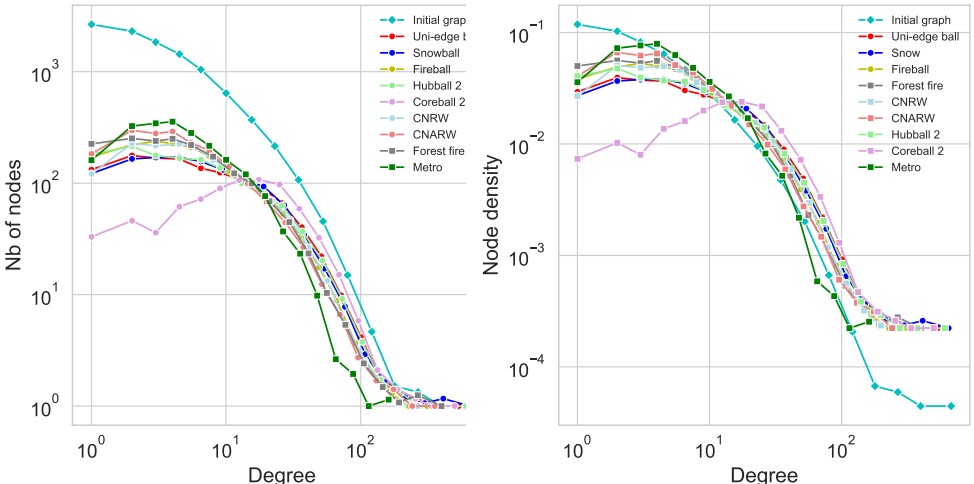

**Figure 4.** Degree distribution (**left**) and density (**right**) of the collected nodes on the Facebook graph. The sampling reduces the graph to 20% of its original size. The curve corresponding to the initial graph is in cyan.

In order to further compare the different graph exploration methods, several metrics have been computed, using different real-world datasets taken from [19,21]. The main characteristics of those datasets have been summarized in Table 1. Each graph has been sampled to get 10% of its nodes (and 20% for the networks having less than 50 k nodes). For each sampled graph, its degree distribution is compared to the original one using the Kolmogorov-Smirnov (KS) test, in Table 2. In order to better exhibit the behavior on high-degree nodes, the KS test is also performed on a partial degree distribution, for degrees higher than the mean degree in the original graph. Other parameters, namely, the average clustering coefficient relative error, transitivity ratio relative error, average PageRank fraction from the original graph, sampled edges ratio and density, are compared in Tables 3–6. On these tables, Coreball refers to Coreball 2 and Hubball to Hubball 2.

**Table 1.** Datasets used in numerical experiments. Only the largest connected component has been kept in case of a disconnected graph.

|  | $|V|$ | $|E|$ |
|---|---|---|
| Facebook [20] | 22 k | 171 k |
| Github [20] | 37.7 k | 289 k |
| Google+ [21] | 202 k | 1.13 M |
| Youtube [21] | 495 k | 1.93 M |

**Table 2.** Kolmogorov-Smirnov test for degree distribution similarity (lower is closer) for the Facebook and Github graphs sampled at 20%, Google+ and Youtube graphs sampled at 10%. The degree distribution is considered either fully ("full" column) or for high-degree (greater than the mean degree of all nodes in the graph) nodes only.

|  | Facebook (20%) | | Github (20%) | | Google+ (10%) | | Youtube (10%) | |
|---|---|---|---|---|---|---|---|---|
|  | full | $d >$ mean | full | $d >$ mean | full | $d >$ mean | full | $d >$ mean |
| Metropolis-Hastings | 0.118 | 0.127 | 0.174 | 0.059 | 0.216 | 0.105 | 0.237 | 0.020 |
| CNRW | 0.227 | 0.053 | 0.278 | 0.076 | 0.591 | 0.153 | 0.362 | 0.178 |
| CNARW | 0.181 | 0.038 | 0.273 | 0.065 | 0.489 | 0.161 | 0.353 | 0.141 |
| Forest Fire | 0.160 | 0.016 | 0.281 | 0.059 | 0.529 | 0.102 | 0.318 | 0.141 |
| **Fireball** | 0.266 | 0.077 | 0.306 | 0.077 | 0.557 | 0.113 | 0.396 | 0.169 |
| **Edgeball** | 0.268 | 0.084 | 0.211 | 0.035 | 0.638 | 0.164 | 0.342 | 0.169 |
| **Hubball** | 0.243 | 0.096 | 0.112 | 0.027 | 0.573 | 0.140 | 0.279 | 0.099 |
| **Coreball** | 0.537 | 0.183 | 0.520 | 0.056 | 0.702 | 0.188 | 0.499 | 0.223 |
| Snowball | 0.224 | 0.082 | 0.073 | 0.022 | 0.482 | 0.052 | 0.279 | 0.125 |

**Table 3.** Numerical results for the Facebook graph sampled at 10% and 20%. Full graph's average clustering coefficient is 0.36, transitivity ratio is 0.23, density is $6.7 \times 10^{-4}$. Relative error for the average clustering coefficient and transitivity ratio is shown between parentheses.

| Sampling Method | | Avg. Clustering Coef. | Transitivity Ratio | PageRank Ratio | Density |
|---|---|---|---|---|---|
| Metropolis-Hastings | 10% | 0.420 (16.83%) | 0.405 (76.25%) | 1.490 | $4.91 \times 10^{-3}$ |
| | 20% | 0.409 (12.57%) | 0.330 (42.15%) | 1.463 | $3.08 \times 10^{-3}$ |
| CNRW | 10% | 0.464 (29.06%) | 0.337 (45.15%) | 2.007 | $9.52 \times 10^{-3}$ |
| | 20% | 0.445 (23.78%) | 0.299 (28.66%) | 1.771 | $5.22 \times 10^{-3}$ |
| CNARW | 10% | 0.369 (2.49%) | 0.287 (23.42%) | 2.057 | $7.99 \times 10^{-3}$ |
| | 20% | 0.375 (4.19%) | 0.274 (17.77%) | 1.757 | $4.75 \times 10^{-3}$ |
| Forest fire | 10% | 0.320 (11.10%) | 0.291 (25.23%) | 1.773 | $8.49 \times 10^{-3}$ |
| | 20% | 0.411 (14.37%) | 0.275 (18.42%) | 1.600 | $4.46 \times 10^{-3}$ |
| **Fireball** | 10% | 0.413 (14.82%) | 0.303 (30.32%) | 1.887 | $1.09 \times 10^{-2}$ |
| | 20% | 0.392 (8.95%) | 0.254 (9.12%) | 1.650 | $5.96 \times 10^{-3}$ |
| **Edgeball** | 10% | 0.441 (22.59%) | 0.238 (2.44%) | 1.854 | $1.50 \times 10^{-2}$ |
| | 20% | 0.449 (24.73%) | 0.257 (10.53%) | 1.664 | $6.23 \times 10^{-3}$ |
| **Hubball** | 10% | 0.484 (34.46%) | 0.220 (5.51%) | 1.805 | $1.41 \times 10^{-2}$ |
| | 20% | 0.460 (27.94%) | 0.223 (4.02%) | 1.619 | $6.46 \times 10^{-3}$ |
| **Coreball** | 10% | 0.476 (32.32%) | 0.291 (25.39%) | 2.030 | $1.97 \times 10^{-2}$ |
| | 20% | 0.461 (28.23%) | 0.300 (29.20%) | 1.823 | $7.57 \times 10^{-3}$ |
| Snowball | 10% | 0.373 (3.81%) | 0.245 (5.66%) | 1.750 | $1.22 \times 10^{-2}$ |
| | 20% | 0.399 (10.87%) | 0.243 (4.48%) | 1.660 | $5.69 \times 10^{-3}$ |

**Table 4.** Numerical results for the Github graph sampled at 10% and 20%. Full graph's average clustering coefficient is 0.168, transitivity ratio is $1.24 \times 10^{-2}$, density is $4.07 \times 10^{-4}$. Relative error for the average clustering coefficient and transitivity ratio is shown between parentheses.

| Sampling Method | | Avg. Clustering Coef. | Transitivity Ratio | PageRank Ratio | Density |
|---|---|---|---|---|---|
| Metropolis-Hastings | 10% | 0.110 (34.64%) | 0.056 (355.32%) | 2.070 | $2.56 \times 10^{-3}$ |
| | 20% | 0.166 (1.22%) | 0.032 (160.00%) | 2.096 | $2.18 \times 10^{-3}$ |
| CNRW | 10% | 0.237 (41.31%) | 0.061 (397.52%) | 3.763 | $6.74 \times 10^{-3}$ |
| | 20% | 0.214 (27.73%) | 0.041 (227.87%) | 2.631 | $3.55 \times 10^{-3}$ |
| CNARW | 10% | 0.211 (25.79%) | 0.061 (397.22%) | 3.778 | $6.75 \times 10^{-3}$ |
| | 20% | 0.186 (10.72%) | 0.041 (232.39%) | 2.645 | $3.54 \times 10^{-3}$ |
| Forest fire | 10% | 0.238 (41.91%) | 0.063 (408.04%) | 3.656 | $6.30 \times 10^{-3}$ |
| | 20% | 0.190 (13.41%) | 0.038 (207.28%) | 2.607 | $3.59 \times 10^{-3}$ |
| **Fireball** | 10% | 0.246 (46.53%) | 0.048 (288.58%) | 3.243 | $5.45 \times 10^{-3}$ |
| | 20% | 0.207 (23.26%) | 0.040 (225.95%) | 2.583 | $3.75 \times 10^{-3}$ |
| **Edgeball** | 10% | 0.310 (85.18%) | 0.040 (223.09%) | 3.036 | $4.65 \times 10^{-3}$ |
| | 20% | 0.247 (47.28%) | 0.035 (184.11%) | 2.452 | $3.09 \times 10^{-3}$ |
| **Hubball** | 10% | 0.373 (122.79%) | 0.025 (98.76%) | 2.594 | $3.94 \times 10^{-3}$ |
| | 20% | 0.348 (107.42%) | 0.020 (61.37%) | 2.092 | $2.48 \times 10^{-3}$ |
| **Coreball** | 10% | 0.252 (50.12%) | 0.054 (337.38%) | 4.247 | $1.05 \times 10^{-2}$ |
| | 20% | 0.188 (12.06%) | 0.039 (214.62%) | 2.968 | $4.84 \times 10^{-3}$ |
| Snowball | 10% | 0.435 (159.64%) | 0.023 (86.65%) | 2.987 | $4.28 \times 10^{-3}$ |
| | 20% | 0.393 (134.28%) | 0.019 (49.94%) | 2.027 | $2.48 \times 10^{-3}$ |

**Table 5.** Numerical results for the Google+ graph sampled at 10%. Full graph's average clustering coefficient is 0.148, transitivity ratio is 0.238, density is $5.57 \times 10^{-5}$. Relative error for the average clustering coefficient and transitivity ratio is shown between parentheses.

| Sampling Method | Avg. Clustering Coef. | Transitivity Ratio | PageRank Ratio | Density |
|---|---|---|---|---|
| Metropolis-Hastings | 0.255 (72.46%) | 0.312 (30.50%) | 1.457 | $6.74 \times 10^{-4}$ |
| CNRW | 0.397 (168.32%) | 0.308 (28.85%) | 1.910 | $2.71 \times 10^{-3}$ |
| CNARW | 0.284 (92.07%) | 0.303 (26.89%) | 1.909 | $2.37 \times 10^{-3}$ |
| Forest fire | 0.317 (114.05%) | 0.298 (24.65%) | 1.859 | $2.10 \times 10^{-3}$ |
| **Fireball** | 0.346 (133.83%) | 0.317 (32.72%) | 1.839 | $2.33 \times 10^{-3}$ |
| **Edgeball** | 0.363 (145.33%) | 0.264 (10.37%) | 1.744 | $3.06 \times 10^{-3}$ |
| **Hubball** | 0.397 (168.16%) | 0.264 (10.64%) | 1.545 | $2.79 \times 10^{-3}$ |
| **Coreball** | 0.400 (170.69%) | 0.296 (23.83%) | 1.940 | $3.23 \times 10^{-3}$ |
| Snowball | 0.360 (143.13%) | 0.285 (19.50%) | 1.576 | $1.97 \times 10^{-3}$ |

**Table 6.** Numerical results for the Youtube graph sampled at 10%. Full graph's average clustering coefficient is 0.11, transitivity ratio is $8.8 \times 10^{-3}$, density is $1.57 \times 10^{-5}$. Relative error for the average clustering coefficient and transitivity ratio is shown between parentheses.

| Sampling Method | Avg. Clustering Coef. | Transitivity Ratio | PageRank Ratio | Density |
|---|---|---|---|---|
| Metropolis-Hastings | 0.115 (4.24%) | 0.052 (487.46%) | 2.930 | $2.07 \times 10^{-4}$ |
| CNRW | 0.153 (39.22%) | 0.040 (354.69%) | 4.201 | $4.65 \times 10^{-4}$ |
| CNARW | 0.125 (13.55%) | 0.041 (364.51%) | 4.210 | $4.58 \times 10^{-4}$ |
| Forest fire | 0.146 (32.40%) | 0.035 (299.01%) | 4.050 | $4.05 \times 10^{-4}$ |
| **Fireball** | 0.154 (40.13%) | 0.038 (332.71%) | 3.967 | $4.90 \times 10^{-4}$ |
| **Edgeball** | 0.186 (69.20%) | 0.028 (221.57%) | 3.886 | $4.57 \times 10^{-4}$ |
| **Hubball** | 0.306 (177.98%) | 0.009 (6.58%) | 3.219 | $3.74 \times 10^{-4}$ |
| **Coreball** | 0.143 (30.13%) | 0.038 (331.17%) | 4.526 | $5.91 \times 10^{-4}$ |
| Snowball | 0.294 (167.08%) | 0.009 (3.64%) | 3.620 | $3.81 \times 10^{-4}$ |

We also introduce another metric called *interCommunity VIP score* or *IVIP* score that summarizes better the intent we have, i.e., sampling efficiently the influencers in social networks. First a community

detection algorithm (we used Louvain in our experiments) is run on the initial graph. We then select the largest communities, in order to cover a sufficient fraction of the nodes (80% in our experiments). Let us denote by $C = \{C_0, C_1, ...\}$ the set of the selected communities and by $d_{C_k}$ the sum of the degrees of all nodes belonging to the community $C_k$. In the sampled graph, some of the nodes belonging to these communities might be present. We denote by $C_k^s$ the sampled nodes belonging to $C_k$, and by $d_{C_k^s}$ the sum of the degrees (in the original graph) of nodes in $C_k^s$. For each sampled graph, the IVIP score is:

$$\frac{\sum_k d_{C_k^s}}{\sum_k d_{C_k}}.$$

This metric will be higher if the high degree nodes of the large communities are sampled, which is the desired behavior when trying to find influencers in a social network. In order to ensure the stability of IVIP, we computed it for 10 different sampling runs (using different initial seeds) and averaged the results. The results are shown on Table 7.

**Table 7.** IVIP score (higher is better) for the different datasets, averaged over 10 sampling runs for each dataset (standard deviation is shown between parentheses).

| | Facebook | | Github | | Google + | Youtube |
|---|---|---|---|---|---|---|
| | **10%** | **20%** | **10%** | **20%** | **10%** | **10%** |
| Metropolis-Hastings | 0.200 (0.022) | 0.378 (0.028) | 0.299 (0.020) | 0.538 (0.019) | 0.291 (0.019) | 0.367 (0.009) |
| CNRW | 0.322 (0.009) | 0.531 (0.013) | 0.513 (0.004) | 0.704 (0.002) | 0.644 (0.004) | 0.573 (0.001) |
| CNARW | 0.325 (0.011) | 0.523 (0.010) | 0.517 (0.004) | 0.709 (0.003) | 0.613 (0.004) | 0.574 (0.001) |
| Forest Fire | 0.272 (0.026) | 0.451 (0.055) | 0.502 (0.009) | 0.698 (0.021) | 0.549 (0.034) | 0.540 (0.033) |
| **Fireball** | 0.278 (0.030) | 0.482 (0.025) | 0.476 (0.028) | 0.685 (0.019) | 0.580 (0.024) | 0.547 (0.022) |
| **Edgeball** | 0.353 (0.010) | 0.552 (0.009) | 0.430 (0.027) | 0.653 (0.011) | 0.658 (0.004) | 0.549 (0.006) |
| **Hubball** | 0.354 (0.010) | 0.557 (0.005) | 0.333 (0.049) | 0.555 (0.031) | 0.630 (0.008) | 0.450 (0.009) |
| **Coreball** | 0.413 (0.013) | 0.613 (0.013) | 0.584 (0.017) | 0.801 (0.005) | 0.698 (0.003) | 0.643 (0.002) |
| Snowball | 0.274 (0.053) | 0.471 (0.035) | 0.387 (0.066) | 0.632 (0.069) | 0.546 (0.052) | 0.449 (0.037) |

**Transitivity:** When a sampling scheme focuses on collecting more high degree nodes, there is a risk to get stuck in a single community, where these nodes are mostly connected to each other and not to outside communities. The experiments show that Spikyballs, as well as other sampling methods, do not have this behavior and explore more than one community (see IVIP score). This is confirmed by the values of the transitivity ratio in Table 4. Indeed, staying within a community would lead to more connections between nodes so more triangles and a higher transitivity ratio. The experiments show that the values for all sampling methods are close, and even the ones of the Spikyball are slightly smaller.

**Pagerank ratio:** Concerning the pagerank measure, all samplings are increasing the values (ratio >1) of the initial graph. This is expected as it is difficult to collect weakly connected nodes (hence with low pagerank), at the periphery of the graph. We notice a higher value for the Coreball on all graphs. This confirms the fact that it collects more high degree nodes, more central, with high page rank. The other members of the Spikyball have diverse values, showing that the average pagerank of a sampled graph can be controlled by the Spikyball parameters. Coreball focuses on the central part of the network, while Hubball samples a more balanced proportion of central and peripheral nodes.

**Density:** Coreball creates the network with the highest density, confirming again, its tendency to collect high degree nodes. The rest of the Spikyball family have values similar to Forest Fire, CNARW, and CNRW. The standard Snowball provides a network with a lower density, which may depend on the graph and on the starting point of the collection. Metropolis-Hastings gives always the smallest density by far, confirming the experiments on the degree distribution where it collects less highly connected nodes than the other methods.

**IVIP and degree distribution:** As displayed in Table 7, the snowball-sampled graphs achieve poor IVIP scores as they contain less communities than the original graphs. Snowball sampling collects every node around its starting location, preventing it to explore a larger region of the graph.

Metropolis-Hastings, with it tendency to sample more weakly connected nodes, misses a part of the high degree nodes. The Coreball gives the highest scores, showing that it focuses on collecting high degree nodes while at the same time exploring most of the communities. The other members of the Spikyball family obtain intermediate scores. Although degree distribution is better approximated by non-Spikyball sampling schemes, as shown in Table 2, this is done at the expense of having a lower fraction of the influencer nodes in the sampled graph.

*5.2. Influence of the Spikyball Parameters*

We now want to see how the parameters of the Spikyballs influence the sampling of the original network. We analyze the degree distribution of the nodes for different parameters. We adopt two approaches: (1) we plot the degree distribution of the sampled graph, as in the previous subsection, and (2) we plot the degree distribution of the nodes captured by the sampling, using their degree *in the original graph*. These are two different pictures, the sampled nodes have a different degree in the sampled graph and in the original graph (except for the snowball sampling). This is illustrated in Figure 1 where the purple sampled nodes have different number of edges in the two graphs, purple ones in the sampled graphs, purple and black ones in the original graph. The results of the samplings on the Facebook graph can be seen in Figure 4.

We remind that the Hubball exploration is directed by the degree of the nodes at each layer, collecting more neighbors from nodes having a larger (resp. smaller) degree when $\alpha$ is positive (resp. negative). As shown on (c), $\alpha$s with negative values increase the collection of small degree nodes, showing that, on this graph, small degree nodes tend to have more connections to other small degree nodes than to hubs. Positive $\alpha$s have no impact on the distribution compared to $\alpha = 0$ (the Uni-edge ball): following the edges going out of the hubs at layer $k$ does not lead to a higher number of high degree nodes collected. Additional experiment with $\alpha > 2$ (not shown) confirms this behavior. This sampling difference is much smaller in the sampled graph. There is barely any visible impact on the degree distribution of the sampled graph obtained from the Hubball family (a), including Forest Fire and the Snowball.

The effect of the parameters is much more visible in the case of the coreballs (b) and (d). As can be seen, positive $\gamma$ are favoring the collection of high degree nodes. There is more than an order of magnitude difference between $\gamma = -2$ and $\gamma = 2$ in the sampling of weakly connected nodes. The difference is larger when looking at the distribution of degrees in the original graph. As in the case of the Hubballs, sampling differences are attenuated on the final sampled graph.

*5.3. Probability to Visit Influencers*

Since the exploration involves a random part, it is important to know the probability to collect (or miss) the important nodes in the network or in some region of the network. To estimate this probability, we performed 10 successive independent explorations for the Hubball, Coreball, Snowball, CNRW, CNARW, Metropolis-Hastings and Forest fire. For each exploration, the initial starting point was a set of 2 nodes selected randomly inside the network. We then counted the number of times a node was collected over the 10 runs. The results are shown in Figure 5. Ideally, important nodes would be collected at each run (10 times). The curves show that the number of visits depends on the degree of the node. Naturally, highly connected nodes have higher chances to be visited and the number of visits rises with the degree for all samplings. The number of visits is influenced by the number of layers (comparing (a) and (b)), with a better capture of high degree nodes with more layers. This might at first be counterintuitive as with each layer new regions of the graph are explored and the number of possible nodes to visit increases. However, social networks have a small-world property and the exploration can not go very far from the initial node. Typically, the diameter of such networks is around 6, therefore adding more layers allows coming back to the initial node and collecting its neighbors. The diameter of the Facebook graph used for the experiments is 15.

Some samplings perform much better than others. The best sampling approaches for sampling hubs are Uni-edge ball (Hubball 0) and Hubball 2. They are able to collect nodes with degree above 100 with 100% probability. With enough layers, it shows that social network influencers can be captured efficiently by these methods even when the network is sampled randomly, selecting only 10% of the neighbors at each layer.

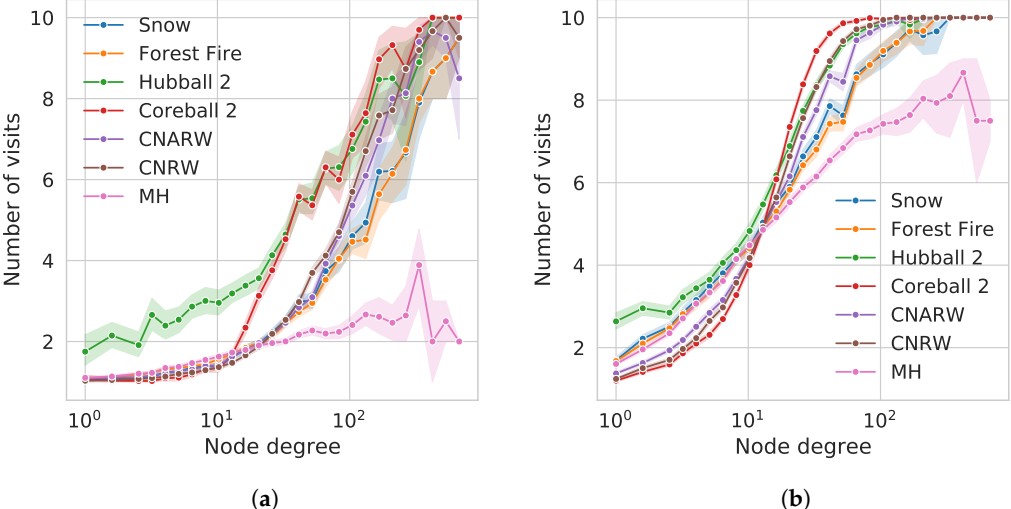

(**a**)                    (**b**)

**Figure 5.** Probability to visit a node with respect to its degree on the Facebook graph for several sampling schemes. Fireball and Edgeball methods have been omitted from the graphs for clarity. The experiment starts using four randomly-selected initial nodes, and is repeated 10 times. The Snowball result is plotted as an indicator of the value for the non-random case, with probability 1. Note that for the Snowball, although the propagation is non-random, the initial starting points are randomized between each run. (**a**) Ball with 2000 nodes (approx. 4 layers of Spikyball) and a selection of 10% of the nodes at each layer (**b**) Ball with 8000 nodes (approx. 8 layers) and the same selection rate. On the left, most of the nodes were visited only once during the 10 tests except for the highest degree ones (above 100 connections). The large 95% confidence interval indicates high variations in the results. On the right, The curve shows a more robust outcome with lower fluctuations and high probability to be visited when a node has a degree larger than 50 (more than 8 visits out of 10). Spikyball-based methods behave in a stabler manner than alternative sampling schemes.

## 6. Discussion

On random graphs, the theoretical and experimental results show that the Snowball, Forest fire and Spikyball explorations methods are very similar. They are based on the same principle. We can expect the same sampling quality while having the option to slightly bend the degree distribution with the Coreball: positive values of $\gamma$ lead to the collection of more high degree nodes and less weakly connected nodes.

In real social networks, the difference in sampling among the Spikyball variants is much more visible. The connections between nodes do not follow simple rules as in the case of synthetic random networks. The degree of a node often has an influence on the degree of its neighbors. Indeed, the change in degree distribution for different parameters of the Hubball family reveals, through Theorem 2, a relationship between the degree of neighbor nodes. Combining the results of Theorem 2 and Section 5.1 (showing that negative $\alpha$ implies more small degree nodes sampled), we can say that nodes with a small degree tend to be connected more frequently to other small degree nodes in the Facebook network. High degree nodes tend to be equally connected to high degree and small degree nodes. This reveal the absence of a hierarchy, like a "rich club" where high degree nodes would connect preferentially with high degree nodes. Among the possible explanations, the way the Facebook

platform is designed does not influence the friendship between active users. Users will connect to their friends in real life independently of their activity in the social network.

The variety of the results obtained for the Spikyball family demonstrates that it is a convenient tool for shaping the sampling distribution. A few parameters control the ability to collect high degree or central nodes or a more balanced mix with the sampling of more peripheral nodes. All the samplings have a good exploration behavior, visiting as many communities as the other state-of-the-art sampling methods.

Concerning the Coreball, the parameter $\gamma$ has a clear impact on the distribution with high values of $\gamma$ favoring the collection of high degree nodes and hubs. This is demonstrated both by the theoretical results and the experiments on degree distribution, IVIP score, PageRank ratio, and density. These results also show that explorations based on a sampling that takes into account the number of connections the neighbors have with the nodes at layer $k$, without knowing their exact degree, lead to an efficient compromise. This partial degree estimation done with the Coreballs does not require to query the exact node degree. From the results, it is a good proxy for a node real degree. It avoids an expensive increase in the number of requests to the social network API.

When exploring a social network, it is desirable to visit and sample less weakly connected nodes. These nodes are so numerous that they mask the important activity without contributing much to the sampled data. In this context, Coreball 2 is the best sampling strategy as the number of sampled weakly connected nodes is decreased by an order of magnitude compared to the standard Snowball or Forest Fire.

The standard graph sampling focuses on a faithful representation of the initial graph. In that case, a random sampling is good as long as it preserves the global graph properties. However, for some applications, it may be necessary to sample key nodes, which makes the random sampling approach ineffective because some of these key nodes may be missed by the random collection process. The experiments show that the Coreball 2 is able to capture hubs and high degree nodes that correspond to key nodes in social networks with a high probability. Hence, Coreball 2 proves to be a robust sampling approach that focuses on the central part of a graph, which can be useful in other applications beyond social networks.

## 7. Conclusions

The Spikyball is a generalization of several exploration sampling schemes. The analysis of its properties, in particular the distribution of degrees of the collected nodes, sheds more light on these approaches. Notably, any of these methods will lead to an equivalent sampling on synthetic random networks. However, when sampling a real network, different approaches may lead to significant discrepancies among the resulting sampled networks.

Its flexibility allows shaping the degree distribution of the sampled graph in a simple manner for a wide range of applications. Depending on the focus of the research, parameters can be chosen to analyze weakly connected nodes, to obtain a high fidelity sub-sampling of a graph or to study the hubs and influencers in social networks. Potential applications of the Spikyball go beyond the scope of social networks, to any large graph where explorations are difficult due to the overwhelming number of nodes and edges. In particular, the Coreball family is able to sample efficiently the "core" of a graph containing its highly connected nodes.

In addition, this promising approach opens new research directions for the analysis and characterization of real-world attributed networks. The Spikyball is presented as a general framework where the exploration rules can be redefined depending on the application. Instead of choosing the degree as the measure of the importance of a node, one could use attributes associated to the nodes. The exploration would be guided by these attributes, revealing new information from the combination of the graph structure and attributes.

**Author Contributions:** Conceptualization, B.R., V.M. and N.A.; Methodology, B.R., V.M. and N.A.; Software, B.R. and N.A.; Validation, B.R., V.M. and N.A.; Writing—original draft preparation, B.R.; Writing—review and editing, B.R., V.M. and N.A.; Visualization, B.R., V.M. and N.A. All authors have read and agreed to the published version of the manuscript.

**Funding:** The authors acknowledges support from the Initiative for Media Innovation.

**Conflicts of Interest:** The authors declare no conflict of interest.

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
