# Peer review of "Spikyball Sampling: Exploring Large Networks via an Inhomogeneous Filtered Diffusion"

_algorithms, doi:10.3390/a13110275_

Round 1

Reviewer 1 Report

Dear authors,

First of all, I would like to congratulate to the authors for this work. Absolutely, it is an interesting topic. Authors presents a new approach for large scale networks that is able to automatically sample relevant parts of a network. In terms of quality of the structure, design and part of the content, this work needs to be improved. In my concern, it still has some major revisions. All my suggestions and comments are focused on a constructive way in order to improve the quality of your work :

  • This technique is a generalization and extension of the snowball sampling method called “Estimating network properties from snowball sampled data” in 2012. In the introduction section, as a reader that is not an expert in this topic can be lost. I missed an introduction of the meaning of Spikyball term. For example, in line 37 authors cited “the Spikyball 1”. In my opinion, the introduction part should be improved. It is important to link the problem with the solution in order to demonstrate the importance of your contribution. In my opinion is not clear.
  • Related work section is excellent. Maybe, it is possible to link the introduction and related work in some manner to emphaticize your work to help the readers to visualize more easily your contribution.
  • The structure of section 3 is not clear. I am agree to explain the main objective of the proposed method. However, all the sub-sections must be cited in one paragraph to understand all the steps of the implementation and mathematical part. Then, it will be more easy to understand section 3.1 and 3.2.
  • In the same manner, the connection between section 3 and 4 is required.
  • A little typing mistake in line 254 and 296 are detected ( I am not sure if it is caused by the compilation project).
  • I am not able to analyse and verify all the mathematical part. I want to congratulate the effort and the description of the theoretical properties section.
  • In section 5 (experimental evaluation), before to start directly with the next sub-section a little paragraph explaining the main validation strategy to validate your results is required.
  • Finally, I think that in this case, mix discussion and conclusion is not a good idea. Maybe, discussion part can be added in the section 5 with the experimental validation. Moreover, all the results can be showed adding all the discussion part. It is an option, but I feel that a main discussion part to understand the good performance and the limitations of the proposed method is required.

Best regards.

Reviewer 2 Report

Graph sampling is indeed an important topic that has received less attention in recent years. This work provides a generalization while connecting Snowball, Forest Fire, and Graph-Expander sampling techniques. It is believed this is especially useful for the social network domain, which their method appears to be motivated from and inspiring the design. 

It was nice to see the authors adding comments throughout the paper of potential adjustments to the sampling strategies, such as for attributed graphs (which are becoming more popular), e.g., in a social network perhaps wanting to pick edges from highly active users.

Although the authors present sufficient related work section, it is good to see section 3.2 that discusses further the connection to these existing sampling methods. This helps solidify the significance of the work from the perspective of prior works. 

One comment for improvement could be adding more experimental components. Although the methods and theory are presented in detail when getting to section 5 it felt a bit lacking. For example, as mentioned in the related work, prior works have focused on many measures, such as clustering coefficient, spectral properties, etc. Indeed on social networks measures such as the influencers/spreaders, but this is somehow already related to properties such as centrality. It would be better to investigate other properties as discussed in the related work that prior works have taken as common for evaluating different sampling methods. Note that overall I believe this is a good work, but believe that further empirical analysis should be included before publication. 

Reviewer 3 Report

This paper aims to solve the problem of sampling from static graphs with massive sizes. The authors have clearly stated the necessity of studying the sampling strategy, the definition of the problem, the model for sampling. The authors have conducted a set of experiments over a number of real-world datasets with different properties, and the experiments results have partly verified the assumptions in the paper.

However, I have the following concerns:

  1. The authors compare Spikyball with Snowball, Fire, Graph Expander and Metropolis Hasting sampling implemented in a open-source sampling toolbox, but these methods are relatively old. It is very necessary to compare some state-of-the-art sampling methods in your experiment. Besides, it is highly recommended that you can put your code link in your experiment because reproducibility is crucial to the research community.
  2. The degree distribution and the corresponding measures of representativeness are inadequate for the new target: how well the “activity” is preserved by the counterpart of samples. Other than the degree distribution, there are a wide range of graph properties that are of importance to investigate the behavior of sampling on the population such as shortest-path, clustering-coefficient and k-core distribution.
  3. The networks for the experiments are not enough. You should add more datasets in different fields and sizes to illustrate the effectiveness of your method.

Round 2

Reviewer 1 Report

Dear authors,

I would like to congratulate to the authors for their efforts to update this second version. All my suggestions and comments were focused on a constructive way in order to improve the quality of this work. It is a pleasure to inform that, after validating your new contributions; your work has been successfully improved. Therefore, the new updated version of this work is ready to be published.

I encourage you to continue working. My best wishes for all of you.

Best regards

Reviewer 3 Report

The author has addressed all the issues that we proposed, I agree to accept it.